

# Systematic analysis and prediction model construction of alternative splicing events in hepatocellular carcinoma: a study on the basis of large-scale spliceseq data from The Cancer Genome Atlas

Lingpeng Yang, Yang He, Zifei Zhang and Wentao Wang

Department of Liver Surgery & Liver Transplantation Center, West China Hospital of Sichuan University, Chengdu, Sichuan, China

## ABSTRACT

Growing evidence showed that alternative splicing (AS) event is significantly related to tumor occurrence and progress. This study was performed to make a systematic analysis of AS events and constructed a robust prediction model of hepatocellular carcinoma (HCC). The clinical information and the genes expression profile data of 335 HCC patients were collected from The Cancer Genome Atlas (TCGA). Information of seven types AS events were collected from the TCGA SpliceSeq database. Overall survival (OS) related AS events and splicing factors (SFs) were identified using univariate Cox regression analysis. The corresponding genes of OS-related AS events were sent for gene network analysis and functional enrichment analysis. Optimal OS-related AS events were selected by LASSO regression to construct prediction model using multivariate Cox regression analysis. Prognostic value of the prediction models were assessed by receiver operating characteristic (ROC) curve and KaplanMeir survival analysis. The relationship between the Percent Spliced In (PSI) value of OS-related AS events and SFs expression were analyzed using Spearman correlation analysis. And the regulation network was generated by Cytoscape. A total of 34,163 AS events were identified, which consist of 3,482 OS-related AS events. UBB, UBE2D3, SF3A1 were the hub genes in the gene network of the top 800 OS-related AS events. The area under the curve (AUC) of the final prediction model based on seven types OS-related AS events was 0.878, 0.843, 0.821 in 1, 3, 5 years, respectively. Upon multivariate analysis, risk score (All) served as the risk factor to independently predict OS for HCC patients. SFs HNRNPH3 and HNRNPL were overexpressed in tumor samples and were signifcantly associated with the OS of HCC patients. The regulation network showed prominent correlation between the expression of SFs and OS-related AS events in HCC patients. The final prediction model performs well in predicting the prognosis of HCC patients. And the findings in this study improve our understanding of the association between AS events and HCC.

Corresponding author
Wentao Wang, wwt0510@163.com

## INTRODUCTION

As the main component of tissues and cells, proteins play a vital role in many life activities. The diversity of proteins contributes to the functional diversity. Alternative splicing (AS) is a significant and ubiquitous post-transcriptional regulatory mechanism that enables eukaryotic cells to generate vast protein diversity based on a limited number of genes (*Baralle & Giudice, 2017*). Genome-wide research indicated that up to 95% of human genes experience some level of AS in physiological processes (*Pan et al., 2008*; *Wang et al., 2008*). Precursor mRNA can be transformed to mature mRNA and further produce versatile protein by removing introns and selectively including or excluding specific exons in human multi-exon genes (*Kelemen et al., 2013*). Recently, growing evidence showed that AS is significantly related to tumor occurrence, progress and therapeutic resistance, and AS is involved in the process of invasion and metastasis of cancer cells (*Wan et al., 2019*; *Oltean & Bates, 2014*). Splicing factor (SF) is the executor of AS events and aberrant expression of SF was associated with oncogenesis process (*Dvinge et al., 2016*).

Liver cancer ranks the 6th place in terms of global tumor incidence, and it is the 4th leading cause of cancer-related death (*Villanueva, 2019*). Hepatocellular carcinoma (HCC), one of the frequently seen primary liver tumors, occupies about 80% of liver cancers. Worldwide, the highest liver cancer morbidity is reported in Asia and Africa. About 75% liver cancers take place in Asian areas, among which China accounts for more than half of the total global cases (*McGlynn, Petrick & London, 2015*). Although great progresses have been achieved in diagnosing and treating liver tumor over past few decades, prognosis for liver tumor remains very poor. Liver cancer has become the second most fatal tumor after pancreatic cancer with a 5-year survival rate of 18% (*Jemal et al., 2017*). In the recent years, some studies have reported the crucial significance of AS in HCC occurrence and development (*Yuan et al., 2017*; *Luo et al., 2017*). However, systematic analysis of the predictive value of AS events in HCC is scarce.

In this study, we collected RNA-seq data, AS events data and corresponding clinical information of 335 HCC patients from TCGA database. Overall survival (OS) related AS events and splicing factors (SFs) were identified. Additionally, a reliable prediction model on the basis of AS events, correlation network between AS events and SFs were constructed.

## MATERIALS AND METHODS

### TCGA-based data collection

The genes expression profile data as well as the clinical information of HCC patients had been collected from TCGA (https://cancergenome.nih.gov/). Information of seven types AS events were collected from the TCGA SpliceSeq database (https://bioinformatics.mdanderson.org/TCGASpliceSeq/), including alternate acceptor site (AA), alternate promoter (AP), alternate donor site (AD), alternate terminator (AT), exon skip (ES), mutually exclusive exons (ME), and retained intron (RI). Due to defining perioperative mortality as death that occurs within 30 days of surgery may underestimate 'true' mortality among patients undergoing hepatic resection (*Mayo et al., 2011*). We only

included patients with a survival time more than 90 days, finally, 335 HCC patients were selected in this study. The Percent Spliced In (PSI) value, rating from 0 to 1 which is used to quantify AS events in general. To generate a reliable set of AS events, we applied a series of stringent filters (Percentage of samples with PSI value ≥75%, standard deviation of PSI value >0.1 and average PSI value > 0.05).

Every AS event was assigned a unique identifier by combining the gene symbol, the ID number in the SpliceSeq database and splicing type. For instance, in the identifier term "CSAD-21952-ES", the gene symbol is CSAD, ID number is 21,952 and splicing type is ES.

## Identification of overall survival related AS events, functional enrichment analysis and gene network construction

We identified overall survival (OS) related AS events using univariate Cox regression analysis with a *P* value <0.05. Upset plot which is similar to a Venn diagram, was introduced to depict the intersections between the seven types of AS events in HCC (*Khan & Mathelier, 2017*). The corresponding genes of OS-related AS events were sent for functional enrichment analysis. Gene Ontology (GO) and Kyoto Encyclopedia of Genes and Genomes (KEGG) pathways with *P* value <0.05 and false discovery rate (FDR) <0.05 were considered significantly. The top significant pathways in KEGG and GO were showed with bar plots. Gene network analysis was performed by inputting the corresponding genes of 800 most significant OS-related AS events to Cytoscape and Reactome FI (version 3.7.1), further, hub genes were selected at the same time.

## Construction of the prediction model based on AS events

Least absolute shrinkage and selection operator (LASSO) regression is suitable for the reduction of high-dimensional data to avoid model overfitting (*Sauerbrei, Royston & Binder, 2007*). When we performed LASSO regression, some variables were eliminated through penalty rules and potential predictors with non-zero coefficients were finally leaved (*Gao, Kwan & Shi, 2010*). We determined the penalty parameter lambda by the cross-validation using the glmnet package. The optimal lambda value corresponding to the minimum value of the cross-validation error mean was identified to determine the potential OS-related AS events (*Tibshirani, 1997*). We selected the optimal OS-related AS events in each type respectively with nonzero coefficients in the LASSO regression, and constructed the prediction models by using multivariate Cox regression analysis. The final prediction model (all types) was constructed by combing seven types of AS events also selected via LASSO regression.

The prediction model was established according to risk score, and the risk score was calculated by the PSI value of each AS events and the corresponding regression coefficient (lnHR) generated from multivariate Cox regression analysis. The formula is as follows: $Riskscore = \sum_{i}^{n} PSIi * \beta i$, where $\beta$ is the regression coefficient.

## Prognostic value evaluation of the prediction model

The HCC patients were divided into low and high risk groups according to the median value of the risk score and Kaplan–Meier survival analysis was performed to compare the OS rate between the two groups within five years. The *p*-values were computed using
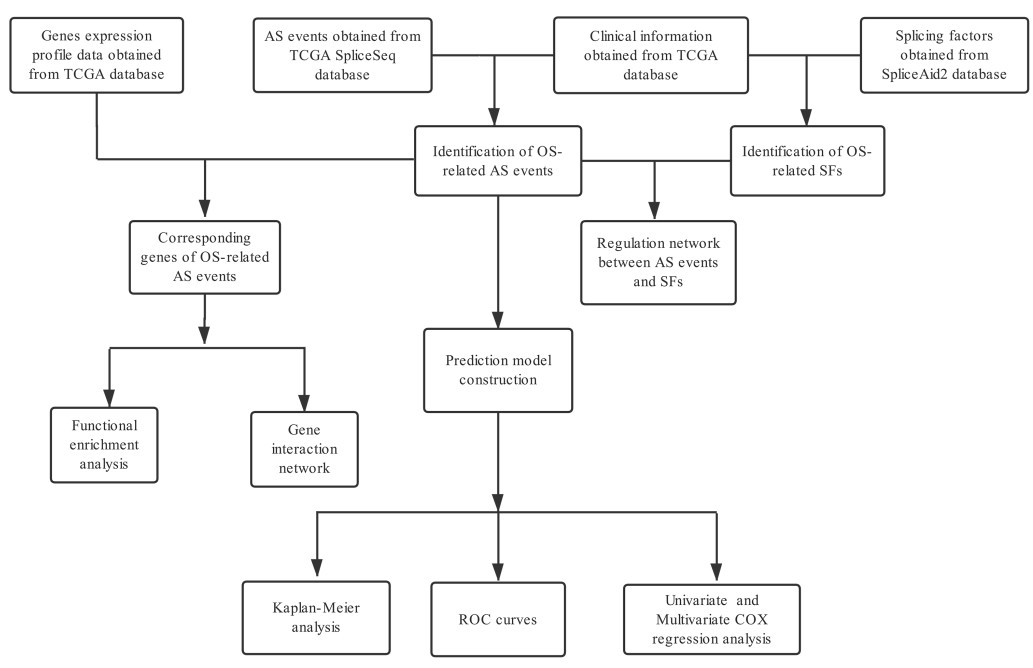

**Figure 1** **Flow chart of the study design.**

log-rank test. Furthermore, ROC curves of 1, 3, 5 years were generated to compare the predictive accuracy of each prediction model.

## Construction of the regulation network between SFs and AS events

SF data was obtained from the SpliceAid2 database (http://www.introni.it/splicing.html). The association of SF expression with OS was analyzed using univariate Cox regression. The relationship between the PSI value of OS-related AS events and SF expression were analyzed using Spearman correlation analysis. Regulation network was generated by Cytoscape and Reactome FI (version 3.7.1).

## Statistical analysis

The R software 3.5.0 was utilized for all statistical analysis. A *P* value <0.05 was deemed to be of statistical significance.

## RESULTS

### Overview of AS events in HCC cohort

A flowchart of our study design is showed in Fig. 1. We detected a total of 34,163 AS events in 8985 genes in 335 HCC patients, comprised of 12,327 ES events in 5,343 genes, 8087 AT events in 3,532 genes, 6,352 AP events in 2,566 genes, 2,666 AA events in 1,937 genes, 2,331 AD events in 1,663 genes, 2,263 RI events in 1,561 genes and 137 ME events in 135 genes (Table 1). We can find that one single gene undergoes up to six types of AS events from the UpSet plot (Fig. 2A). In addition, ES was the most common among seven types of AS events.

**Table 1  Overview of total AS events and OS-related AS events.**

| Type | Total AS events | | OS-related AS events | |
|---|---|---|---|---|
| | AS events | Genes | AS events | Genes |
| AA | 2666 | 1937 | 217 | 203 |
| AD | 2331 | 1663 | 231 | 205 |
| AP | 6352 | 2566 | 643 | 411 |
| AT | 8087 | 3532 | 887 | 534 |
| ES | 12327 | 5343 | 1264 | 988 |
| ME | 137 | 135 | 16 | 16 |
| RI | 2263 | 1561 | 224 | 196 |
| ALL | 34163 | 8985 | 3482 | 2203 |

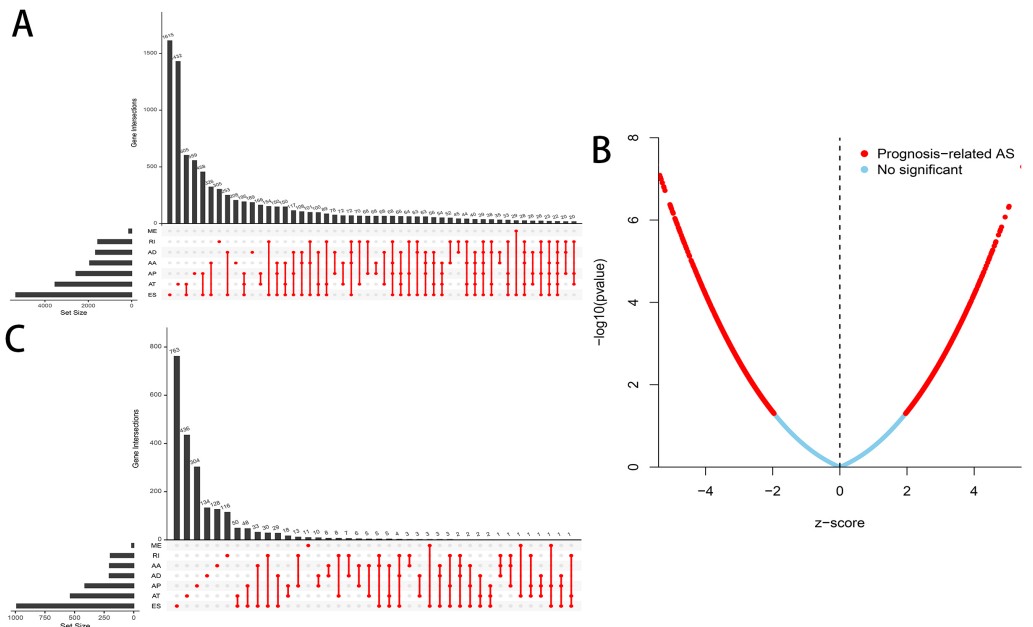

**Figure 2  Overview of AS events in HCC patients.** (A) Upset plot of the intersections between the seven types of AS events. (B) Volcano plot of OS-related AS events (red dot) and OS-irrelated AS events (blue dot). (C) Upset plot of the intersections between the seven types of OS-related AS events.

## The OS-related AS events of HCC

A total of 3,482 OS-related AS events in 2,203 genes were detected using univariate Cox analysis, including 1,264 ES events in 988 genes, 887 AT events in 534 genes, 643 AP events in 411 genes, 217 AA events in 203 genes, 231 AD events in 205 genes, 224 RI events in 196 genes and 16 ME events in 16 genes (Table 1). A volcano plot of these AS events was provided in Fig. 2B. With the display of UpSet plot, one single gene could have up to four types OS-related AS events (Fig. 2C). The top 20 significant OS-related AS events (if available) for each AS type were showed by bubble plots in Figs. 3A–3G. Obviously, most

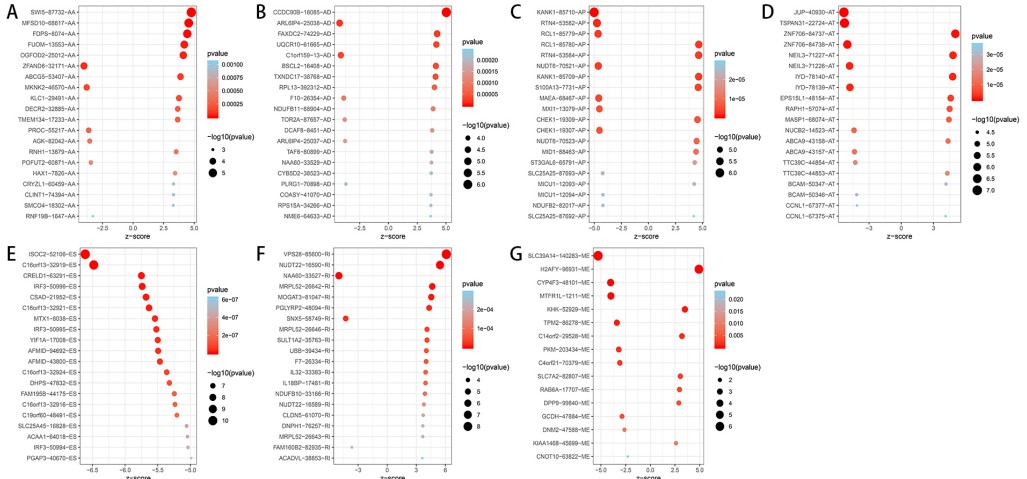

**Figure 3** **Bubble plots of OS-related AS events in HCC patients.** (A–F) The top 20 significant OS-related AS events for AA, AD, AP, AT, ES and RI. (G) 16 OS-related AS events for ME.

of AS events in ES were favorable prognostic elements. However, most of these AS events in RI, AA and AD were adverse prognostic elements.

We established a gene interaction network by sending the corresponding genes of the top 800 most significant OS-related AS events to Cytoscape. The results indicated that UBB, UBE2D3, SF3A1 were the hub genes of this gene network (Fig. 4A). Furthermore, 2,203 genes from 3,482 OS-related AS events were used for KEGG and GO enrichment analysis to explore the pathways and biological functions of the OS-related AS genes. The top 10 significant terms of GO enrichment analysis were presented in Fig. 4B. Such as spliceosomal complex, adherens junction in cellular component (CC); damaged DNA binding, cell adhesion molecule binding in molecular function (MF); protein targeting, actin cytoskeleton reorganization in biological process (BP). A total of nine KEGG pathways were identified, such as base excision repair, pyruvate metabolism and PPAR signaling pathway (Fig. 4C).

## Construction of the prediction model for HCC patients

We used LASSO regression to select the top 10 optimal OS-related AS events (if available) and then construct the prediction model (Fig. 5). Three OS-related AS events were selected for ES; four OS-related AS events for ME; five OS-related AS events for AP and AT; six OS-related AS events for AA, AD and RI; nine OS-related AS events for the final prediction model (Table 2). Risk scores were computed according to the selected AS events, and HCC patients were divided into low and high risk groups on the basis of the median value of risk scores. The distribution of survival status in low and high risk groups, risk score curves and the PSI value heat map of the AS events for eight prediction models were visualized in Fig. 6. The results of Kaplan–Meier survival analysis showed that all of the eight prediction models possess significant ability to predict the prognosis of HCC patients between low and high risk group (Fig. 7). However, according the results of ROC curves, the final prediction

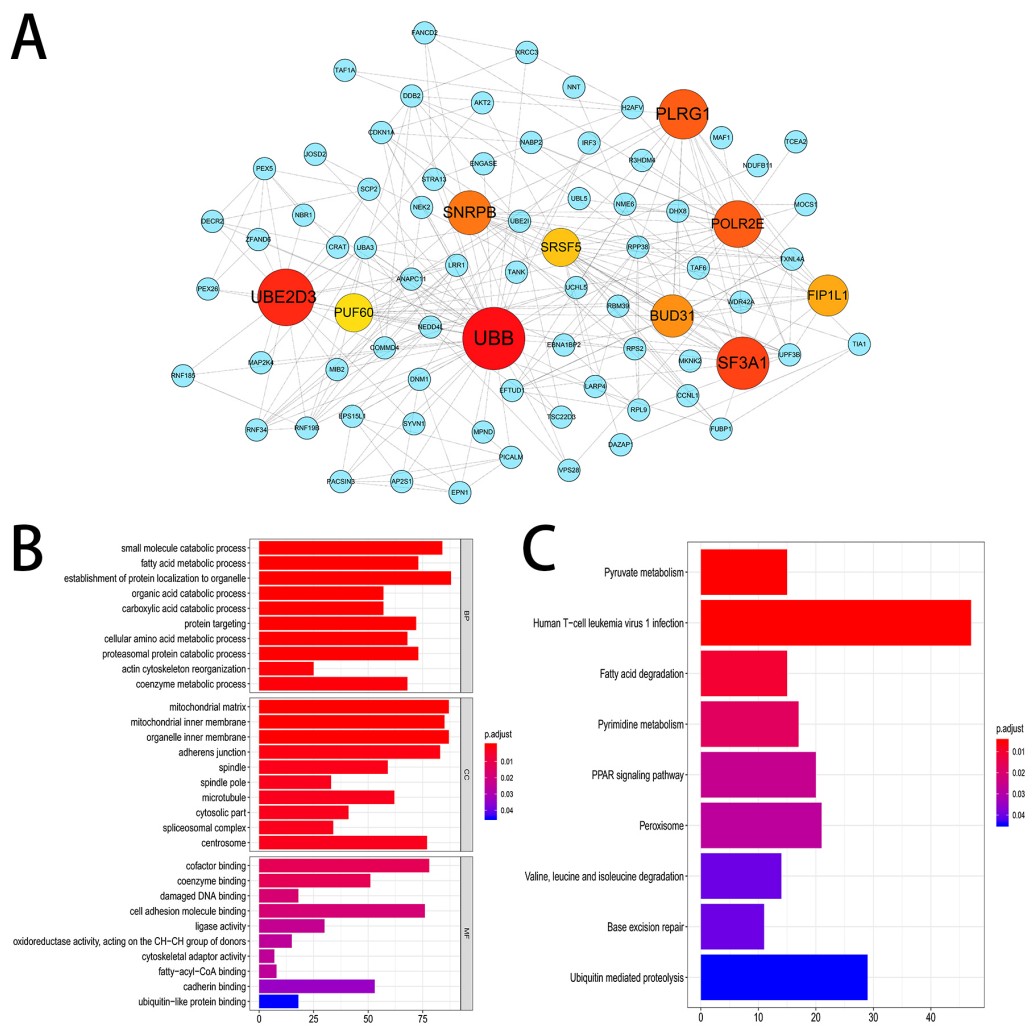

**Figure 4  Gene interaction network and functional enrichment analysis.** (A) Gene interaction network of the top 800 most significant OS-related AS events. (B) The top 10 significant terms of GO enrichment analysis (BP/CC/MF). (C) Nine terms of KEGG enrichment analysis.

model exhibited the most powerful predictive efficiency than other models established by one single AS type with the maximum AUCs of 0.878, 0.843, 0.821 in 1, 3, 5 years ROC curves (Fig. 8).

Moreover, we assessed the prognostic value of risk score and other clinicopathological characteristics using univariate and multivariate Cox regression analysis. Univariate analysis suggested that advanced clinical stage (HR = 2.053, 95% CI [1.553–2.714], $p < 0.001$), high T classification (HR = 1.962, 95% CI [1.515–2.542], $p < 0.001$) and high risk score (All) (HR = 1.105, 95% CI [1.076–1.135], $p < 0.001$) were associated with poor prognosis (Fig. 9A). Upon multivariate analysis, risk score (All) (HR = 1.101, 95% CI [1.069–1.135], $p < 0.001$) served as the risk factor to independently predict OS for HCC patients (Fig. 9B).

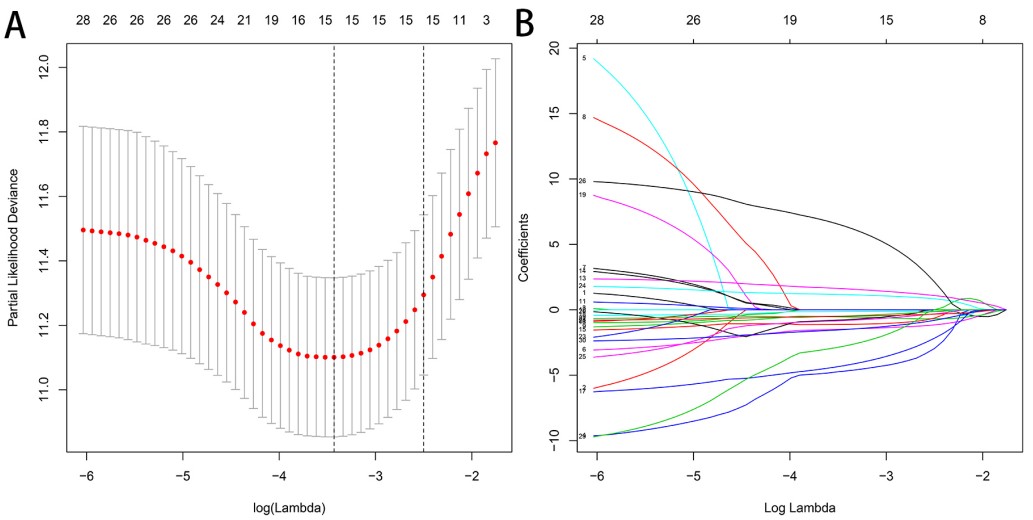

**Figure 5   Selection of the optimal AS-related events used for construction of the final prediction model by LASSO regression.** (A) Selection of optimal parameter (lambda) in the LASSO model, dotted vertical lines were drawn at the optimal values. (B) LASSO coefficient profiles of the nine OS-related AS events with nonzero coefficients determined by the optimal lambda.

## The regulation network of SFs and OS-related AS events

SF is the executor of AS events and aberrant expression of SF was related to oncogenesis process. A total of 71 SFs data were collected from the SpliceAid2 database (http://www.introni.it/splicing.html). We identified 21 SFs associated with OS of HCC patients using univariate analysis. Then Spearman correlation analysis was performed to determine the correlation between the PSI value of OS-related AS events and SF expression. The significant correlations ($|R|{\geq}0.6$, $p < 0.001$) were selected to construct the regulation network (Fig. 10A). The regulation network consists of 29 OS-related AS events, of which 16 were adverse AS events (red dots) and 13 were favorable AS events (green dots), were significantly correlated with the 13 SFs (blue dots). We can find that all of the SFs were correlated with multiple AS events and played opposite roles in regulating different AS events. Similarly, a part of the AS events could be regulated by different SFs. This phenomenon partly explains that the same transcript can produce multiple different splicing events. In addition, we can detect that the adverse AS events were positively correlated with SF expression (red lines), whereas the favorable AS events were negatively correlated with SF expression (green lines).

The top four most significant correlations between SFs and OS-related AS events ($|R|{\geq}0.7$) are shown in Figs. 10D, 10E, 10H and 10I. The top two counterpart SFs were HNRNPL and HNRNPH3, and the expression of HNRNPL and HNRNPH3 in tumor sample were significantly higher than adjacent normal sample (Figs. 10B and 10F). HCC patients were classified into low and high risk groups according to the median value of the two SFs expression. Also, Kaplan–Meier survival analysis showed statistical difference in survival time between the two groups (Figs. 10C and 10G).

**Table 2** Information of AS events used for construction of prediction model.

| Type | ID | Coef | HR | 95% Lower | 95% Up | *P* value |
|------|-----|------|-----|-----------|--------|-----------|
| AA | SWI5-87732-AA | 3.44 | 31.34 | 4.32 | 227.57 | 0.00066043 |
| | MFSD10-68617-AA | 5.53 | 251.81 | 4.79 | 13244.14 | 0.006246456 |
| | FDPS-8074-AA | 1.72 | 5.58 | 0.96 | 32.52 | 0.055702088 |
| | FUOM-13553-AA | 4.50 | 89.61 | 2.18 | 3689.99 | 0.017794651 |
| | ZFAND6-32171-AA | −18.97 | 5.79e−9 | 1.84e−14 | 1.83e−3 | 0.003324932 |
| | ABCG5-53407-AA | 1.45 | 4.24 | 1.54 | 11.67 | 0.005133725 |
| AD | CCDC90B-18085-AD | 2.73 | 15.27 | 2.46 | 94.69 | 0.003404273 |
| | FAXDC2-74229-AD | 3.72 | 41.20 | 1.74 | 973.72 | 0.021200397 |
| | C1orf159-13-AD | −2.02 | 0.13 | 0.04 | 0.42 | 0.000547079 |
| | BSCL2-16408-AD | 6.33 | 560.69 | 9.32 | 33724.29 | 0.002462146 |
| | TXNDC17-38768-AD | 12.15 | 1.89e+5 | 8.72 | 4.12e+9 | 0.017081663 |
| | RPL13-392312-AD | 3.25 | 25.81 | 3.24 | 205.55 | 0.002134468 |
| AP | KANK1-85710-AP | −1.12 | 0.33 | 0.16 | 0.68 | 0.003039732 |
| | NUDT6-70521-AP | −1.40 | 0.25 | 0.09 | 0.71 | 0.009102069 |
| | S100A13-7731-AP | 1.33 | 3.76 | 0.93 | 15.27 | 0.063560786 |
| | MAEA-68467-AP | −11.36 | 1.16e−5 | 1.41e−8 | 0.01 | 0.000914974 |
| | MXI1-13079-AP | −1.25 | 0.29 | 0.09 | 0.89 | 0.030826651 |
| AT | JUP-40930-AT | −4.18 | 0.02 | 8.21e−4 | 0.29 | 0.005110540 |
| | TSPAN31-22724-AT | −5.98 | 2.52e−3 | 9.11e−5 | 0.07 | 0.000412724 |
| | ZNF706-84737-AT | 9.10 | 8974.41 | 116.68 | 690281.40 | 3.99e−5 |
| | NEIL3-71227-AT | 1.45 | 4.26 | 1.78 | 10.19 | 0.001109076 |
| | EPS15L1-48154-AT | 5.79 | 327.41 | 15.76 | 6803.41 | 0.000183173 |
| ES | CRELD1-63291-ES | −13.64 | 1.19e−6 | 5.85e−10 | 2.44e−3 | 0.000452970 |
| | CSAD-21952-ES | −3.21 | 0.04 | 0.01 | 0.19 | 4.24e−5 |
| | IRF3-50995-ES | −2.62 | 0.07 | 0.03 | 0.20 | 6.14e−7 |
| ME | SLC39A14-140283-ME | −1.96 | 0.14 | 0.04 | 0.45 | 0.001037644 |
| | H2AFY-96931-ME | 3.18 | 24.02 | 2.71 | 211.85 | 0.004205834 |
| | CYP4F3-48101-ME | −4.59 | 0.01 | 2.74e−4 | 0.37 | 0.012551164 |
| | MTFR1L-1211-ME | −2.30 | 0.10 | 0.03 | 0.38 | 0.000662517 |
| RI | VPS28-85600-RI | 6.06 | 429.08 | 5.62 | 32758.26 | 0.006135308 |
| | NUDT22-16590-RI | 2.69 | 14.79 | 3.18 | 68.86 | 0.000599344 |
| | NAA60-33527-RI | −1.87 | 0.15 | 0.04 | 0.65 | 0.010841633 |
| | MOGAT3-81047-RI | 5.24 | 188.66 | 5.22 | 6812.78 | 0.004190482 |
| | SNX5-58749-RI | −3.94 | 0.02 | 3.24e−3 | 0.12 | 1.70e−5 |
| | UBB-39434-RI | 44.68 | 2.54e+19 | 2.40e+10 | 2.69e+28 | 2.50e−5 |
| | CSAD-21952-ES | −2.15 | 0.12 | 0.02 | 0.58 | 0.008527832 |
| | NUDT22-16590-RI | 2.26 | 9.58 | 2.24 | 40.90 | 0.00228534 |
| | TSPAN31-22724-AT | −6.17 | 2.08e−3 | 4.74e−5 | 0.09 | 0.001376182 |
| | CCDC90B-18085-AD | 1.53 | 4.62 | 0.68 | 31.15 | 0.116422196 |

**Table 2** (*continued*)

| Type | ID | Coef | HR | 95% Lower | 95% Up | *P* value |
|------|-----|------|-----|-----------|--------|-----------|
| ALL | IRF3-50994-ES | −1.71 | 0.18 | 0.03 | 1.02 | 0.053074975 |
| | ZNF706-84737-AT | 8.94 | 7608.05 | 76.67 | 754961.11 | 0.000138991 |
| | KANK1-85710-AP | −0.80 | 0.45 | 0.22 | 0.94 | 0.033403 |
| | PGAP3-40670-ES | −6.81 | 1.11e−3 | 4.06e−7 | 3.02 | 0.091751791 |
| | NAA60-33527-RI | −2.27 | 0.10 | 0.03 | 0.41 | 0.001319454 |

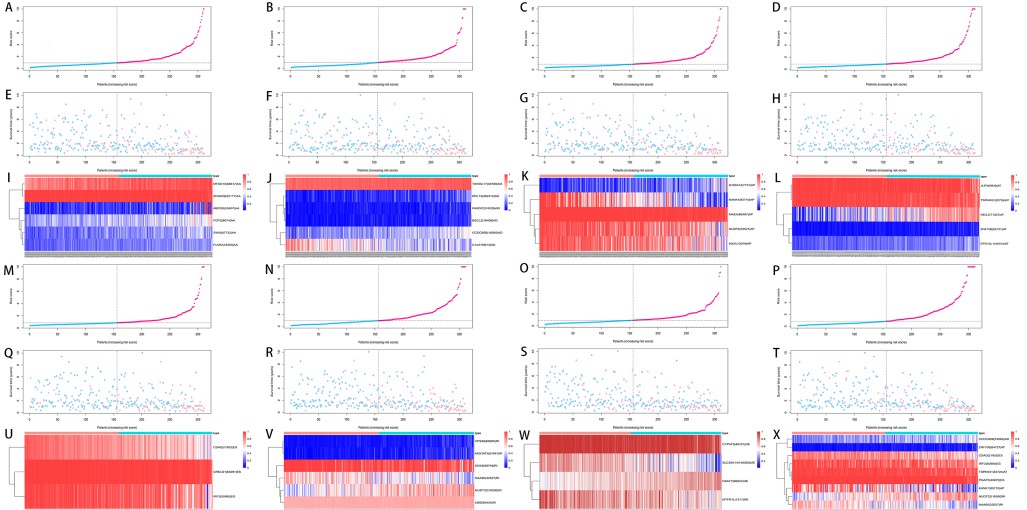

**Figure 6** **Analysis of the prediction models in HCC patients.** HCC patients were divided into low and high risk groups on the basis of the median value of risk scores. (A–D, M–P) The risk score curves for AA, AD, AP, AT, ES, RI, ME events and all types of AS events. (E–H, Q–T) Survival status and survival times of HCC patients ranked by risk score for AA, AD, AP, AT, ES, RI, ME events and all types of AS events. (I–L, U–X) The PSI value heatmap of the AS events for AA, AD, AP, AT, ES, RI, ME events and all types of AS events. Color from blue to red indicates the increasing PSI score of corresponding AS event from low to high.

# DISCUSSION

AS is a significant regulatory process for generating protein isoforms with a variety of functional characteristics. Abnormality of AS events are closely associated with tumor occurrence, development and metastasis (*Oltean & Bates, 2014*; *Liu & Cheng, 2013*; *Spaethling et al., 2016*). In recent decades, with the rapid development of high-throughput sequencing technology, the potential significance of AS events in malignant tumor has achieved great advancement. However, there are few studies focus on the systematic analysis of AS events in HCC patients.

Then we screened OS-related AS events and OS-related SFs in HCC patients via the analysis of TCGA and SpliceAid2 database. A total of 3,482 OS-related AS events in 2,203 genes and 21 OS-related SFs were detected using univariate Cox analysis, which shows that AS events are ubiquitous in HCC patients and limited SFs can regulate massive

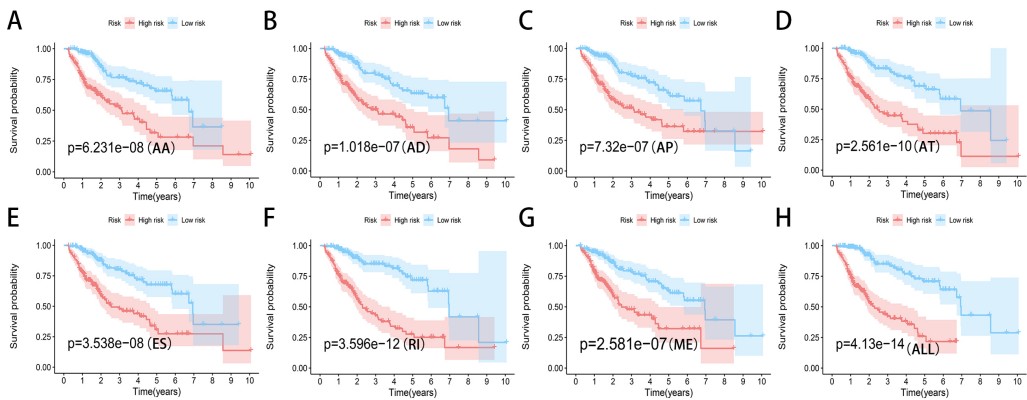

**Figure 7** **Kaplan-Meier plots of the eight prediction models constructed with AS events for HCC patients.** (A–H) Kaplan-Meier plots of prediction models constructed with AA, AD, AP, AT, ES, RI, ME events and all types of AS events.

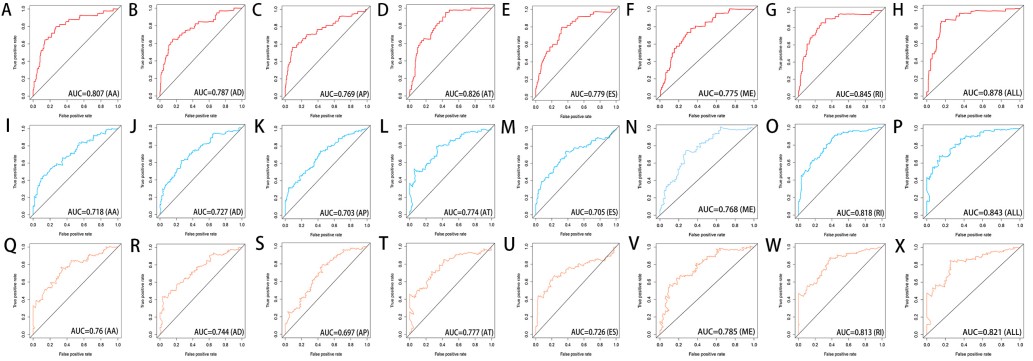

**Figure 8** **ROC curves of the eight prediction models for risk prediction in 1 year, 3 years and 5 years.** ROC curves in 1 year (A–H), 3 years (I–P) and 5 years (Q–X) for AA, AD, AP, AT, ES, ME, RI models and the final model (All).

AS. The gene interaction network was established using Cytoscape based on the top 800 OS-related AS events, and UBB, UBE2D3, SF3A1 were identified as the hub genes of this gene network. Ubiquitin is a small and highly conserved protein expressed in all eukaryotic cells. Over expression of ubiquitin B (UBB) was reported in non-small cell lung cancer and cervical cancer, UBB may serve as a potential therapy and prevention target (*Tang et al., 2015*; *Tian et al., 2013*). Nevertheless, the relationship between HCC and UBB is not clear. Ubiquitin-conjugating enzyme E2D3 (UBE2D3) is a member of the E2 family, which is involved in the ubiquitin proteasome pathway to regulate the basic activities of cells, such as DNA damage response, cell cycle control, apoptosis, and tumorgenesis. A previous study demonstrated that UBE2D3 plays a significant role in the development of esophageal cancer (*Guan et al., 2015*). SF3A1 is a critical spliceosome gene which participated in normal splicing events and spliceosome assembly (*Chen et al., 2015*). SF3A1 has been reported to be related to susceptibility of breast cancer and lung cancer (*Hu et al., 2011*;

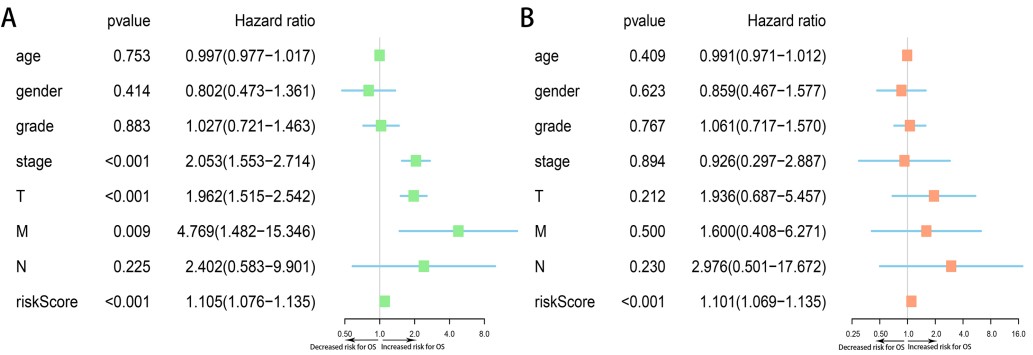

**Figure 9** Forest plots of Cox regression analysis of clinicopathological characteristics and risk score (All). (A) Univariate analysis, (B) multivariate analysis.

*Michailidou et al., 2013*). And the mutation of SF3A1 were involved in some cancers, including esophageal adenocarcinoma, osteosarcomas, ovarian carcinoma and gastric cancer (*Chen et al., 2015*). Additionally, KEGG and GO enrichment analysis were performed and we found that these genes were closely associated with spliceosomal complex, cell adhesion molecule binding, actin cytoskeleton reorganization, base excision repair, etc. From the above mentioned, UBB, UBE2D3 and SF3A1 as the hub genes in this gene interaction network and may be potential targets for the prevention and treatment of HCC in the future.

In the recent years, with the fast development of high-throughput sequencing, massive potential prognosis biomarkers or therapy targets of tumor were emerging, such as mRNA, miRNA, lncRNA, and methylation (*Zhao et al., 2018*; *Ji et al., 2018*; *Cai et al., 2019*; *Wu et al., 2017*). However, the focus of these studies is limited to the transcriptome-level analysis. The prognostic value of AS events has a great potential for development. *Tremblay et al. (2016)* firstly investigated differential AS events between HCC sample and normal liver sample based on TCGA database, and they provided an overview of misregulated AS events in different types of HCC (e.g., HBV-related HCC, HCV-related HCC, HBV&HCV-associated HCC and virus-free HCC). But they did not explore the association between the AS events and the prognosis of HCC patients. Subsequently, some researchers identified OS-related AS events to establish a prediction model in HCC patients (*Chen et al., 2019*; *Zhu et al., 2019*). However, Chen et al. reported that the AUC of ROC curve for the final prediction model constructed with 10 AS events was only 0.752. Zhu et al. constructed the final prediction model with up to 33 AS events, but the AUC of ROC curve was only 0.806. In this study, the final prediction model exhibited the most powerful predictive efficiency than other models established by one single AS type with the maximum AUCs of 0.878, 0.843, 0.821 in 1, 3, 5 years ROC curves. And our final prediction model was constructed with only nine OS-related AS events.

SF plays a significant role in the regulation of AS events, which affects the selection of exons and splicing sites by identifying and combining to cis-regulatory elements of pre-mRNA. Aberrant expression of SF was related to oncogenesis process. The regulation

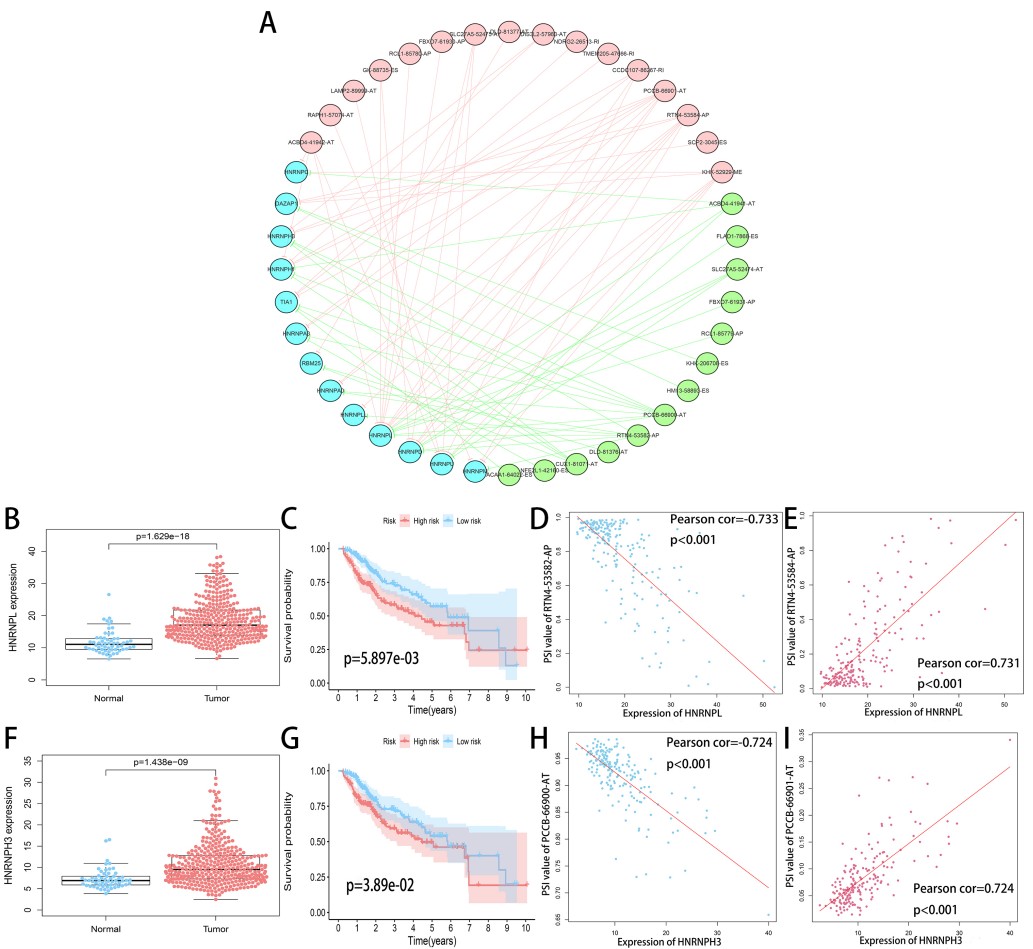

**Figure 10** **Regulation network of SFs and OS-related AS events.** (A) Correlation network between the expression of OS-related SFs and the PSI values of OS-related AS events. The expression of OS-related SFs (blue dots) were positively (red line) or negatively (green line) correlated with the PSI values of OS-related AS events. The favorable AS events are showed by green dots while adverse AS events are showed by red dots. (B, F) Expression of HNRNPL and HNRNPH3 in tumor sample (red dots) and adjacent normal sample (blue dots). (C, G) Kaplan-Meier plots of HNRNPL and HNRNPH3, HCC patients were divided into low (blue curve) and high risk (red curve) groups according to the median value of the two SFs expression. (D, E) Correlation analysis between the expression of HNRNPL and the PSI values of RTN4-53582-AP, RTN4-53584-AP. (H, I) Correlation analysis between the expression of HNRNPH3 and the PSI values of PCCB-66900-AT, PCCB-66901-AT.

network was constructed to show the correlation between the PSI value of OS-related AS events and SF expression. The top two SFs were HNRNPL and HNRNPH3 identified according to the correlation coefficient ($|R| \geq 0.7$). Compared to adjacent normal sample, the expression of HNRNPL and HNRNPH3 were higher in tumor sample. And patients with higher expression of HNRNPL and HNRNPH3 had more dismal prognosis. From the regulation network, we found that the adverse AS events were positively correlated with the expression of HNRNPL and HNRNPH3, whereas the favorable AS events were negatively correlated with the expression of HNRNPL and HNRNPH3, which was consistent with the

results of survival analysis. HNRNPL and HNRNPH3 are members of HNRNPs family. HNRNPs directly modulate the AS of a group of RNAs and serve as multifunctional RNA-binding proteins for mRNA stabilization, transportation and translation (*Fei et al., 2017*). Aberrant expression of HNRNPL and its RNA target are closely associated with the proliferation, invasion and metastasis of tumor cells (*Kedzierska & Piekielko-Witkowska, 2017*; *Geuens, Bouhy & Timmerman, 2016*). A recent study have demonstrated that HNRNPL was highly expressed in HCC samples and down-regulation of HNRNPL expression can significantly inhibit the proliferation and migration of liver cancer cells (*Yau et al., 2013*), which was in accordance with our results. However, to the best knowledge of us, there is no research report the actual regulatory mechanism between the two prognostic SFs and OS-related AS events, and further elucidation with *in vivo* or *vitro* experiments is urgently needed.

Although the findings in the present study improves our understanding of the association between AS events and HCC, some limitations existed in this study. First, this study was conducted based on the data obtained from one public database with a relatively small sample size. Second, due to the lack of external data, we did not make a cohort verification for the prediction model. Third, we did not perform a functional experiment and could not clearly elucidate the underlying mechanism between AS events and SFs in HCC patients. It is essential to carry out functional experiments and clinical trials with a large sample size of HCC patients to confirm the findings of this study in the future.

## CONCLUSIONS

In summary, we performed a systematic analysis of AS events in HCC and constructed prediction models based on OS-related AS events with well performance in predicting the prognosis of HCC patients. The AS events used for the construction of the final prediction model may be the most significant AS events in exploring the latent mechanism in initiation and development of HCC, have a great potential for clinical application as therapeutic and preventive targets of HCC patients. Furthermore, we established a regulation network between SFs and OS-related AS events. Although the findings in the present study improve our understanding of the association between AS events and HCC to some extent. *In vitro/vivo* function experiments are also urgently needed in the future to understand the mechanism between AS events and SFs in the HCC patients.

### Funding
The current work was supported by the National Natural Science Foundation of China (No. 81770566), the New Medical Technology Foundation of West China Hospital of Sichuan University (No. XJS2016004), and the Science and Technology Program of Sichuan Science and Technology Department (No. 2019YFS0029). The funders had no role in study design, data collection and analysis, decision to publish, or preparation of the manuscript.

## Grant Disclosures

The following grant information was disclosed by the authors:

National Natural Science Foundation of China: 81770566.

New Medical Technology Foundation of West China Hospital of Sichuan University: XJS2016004.

Science and Technology Program of Sichuan Science and Technology Department: 2019YFS0029.

## Competing Interests

The authors declare there are no competing interests.

## Author Contributions

- Lingpeng Yang performed the experiments, prepared figures and/or tables, authored or reviewed drafts of the paper, approved the final draft.
- Yang He and Zifei Zhang analyzed the data, contributed reagents/materials/analysis tools, prepared figures and/or tables, approved the final draft.
- Wentao Wang conceived and designed the experiments, authored or reviewed drafts of the paper, approved the final draft.

## Data Availability

The raw data is available in the Supplemental Files.

## Supplemental Information

Supplemental information for this article can be found online at http://dx.doi.org/10.7717/peerj.8245#supplemental-information.

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
