# Peer review of "Systematic analysis and prediction model construction of alternative splicing events in hepatocellular carcinoma: a study on the basis of large-scale spliceseq data from The Cancer Genome Atlas"

_PeerJ, doi:10.7717/peerj.8245_

## Round 0.1 · original submission · Major Revisions

Please address critiques of all reviewers and amend your manuscript accordingly.

Reviewer 1 ·

Basic reporting

Not all critical literature was cited and commented.

Experimental design

The study is purely theoretical and incomplete. Experimental confirmation of the propossed model is required.

Validity of the findings

I have doubts regarding the originality of essential aspects of this study.

Additional comments

In this study, the authors have used data available at The Cancer Genome Atlas to analyze the existence of patterns of alternative splicing events characteristics of hepatocellular carcinoma (HCC) and have propossed to use them to predict cancer behavior and patients outcome. The study is interesting but, to justify the originality of their work, the authors must clearly define the differences with a similar paper published three years ago (*) that has not even cited or included in the list of references. The valuable results from the present study require confirmation in a validation cohort.

*Global profiling of alternative RNA splicing events provides insights into molecular differences between various types of hepatocellular carcinoma.
Tremblay MP, Armero VE, Allaire A, Boudreault S, Martenon-Brodeur C, Durand M, Lapointe E, Thibault P, Tremblay-L├ętourneau M, Perreault JP, Scott MS, Bisaillon M.
BMC Genomics. 2016 Aug 26;17:683.

·

Basic reporting

Alternative splicing events have been popularly recognized in recent years to contribute towards tumorigenesis. The authors in this study have generated a predictive model to assess patient overall survival in hepatocellular carcinoma based on splicing events. Although, this is not the first study to have used predictive modeling on RNAseq patient database, it is the first to do so more effectively (AUC for ALL events 5 years ROC curve of 0.821). However, there are few points for the authors to consider to make their study clearer and better:

1) Minor grammatical errors throughout the manuscript. Authors are advised to proofread it thoroughly. For example- 'protein' in Lines 37 and 38 should be replaced by 'proteins' and the references in parentheses should come before full stop at the end of a sentence.

2) Figure 5 needs to be in high resolution. It is hard to read the labels for heatmaps.

3) For better understanding of the readers, kindly label the x-axis in Figure 8 well. Is the hazard ratio predicting increase and decrease in overall survival of the patients here?

4) A flowchart describing the process of data processing will be useful for the readers.

Experimental design

1) In the UpSet plot, authors show how different AS events interact, but the predicted model for patient outcomes only uses single or ALL events for OS risk estimate. It will be good to see and understand how and to what extent different AS event types when interacting contribute to the patient outcome in HCC.

2) authors are requested to delineate how the events per AS types were selected for LASSO regression.

Validity of the findings

No comment.

Reviewer 3 ·

Basic reporting

The manuscript passed all the basic reporting requirements

Experimental design

The manuscript passed all the experimental design requirements

Validity of the findings

The manuscript passed all the validity of the findings requirements

Additional comments

Alternative splicing events (ASEs) play a role in cancer development and progression. The authors herein investigated a systematic analysis of ASE and constructed a robust prediction model of hepatocellular carcinoma. The authors claim that the final prediction model performs well in predicting the prognosis of HCC patients and the findings in this study improves understanding of the association between AS events and HCC. I support publishing this manuscript with some major and minor revisions:
1) As the authors mentioned the biggest drawback of the manuscript is lack of experimental verification for understanding the mechanism between AS events and SFs in the HCC patients. Are the authors planning to do any invitro and in vivo assessments as a future work? Can the authors mention in their final conclusions?
2) I recommend the authors to add a flow chart for their study design
3) In the TCGA- based data collection, patients with a survival time less than 90 days were excluded, I assume clinical follow up data can be available? can the authors add a reason why it was excluded for 90 days instead of 30 days or lesser?

---

## Round 0.2 · accepted · Accept

Although one of the reviewers not only remained skeptical, but even worsened their recommendation, I decided to accept this manuscript. This is because I disagree with the "lack of originality" and "study is preliminary" statements.

Reviewer 1 ·

Basic reporting

No additional comments

Experimental design

No additional comments

Validity of the findings

No additional comments

Additional comments

The authors have responded to my comments in a nonconvincing way.

Reviewer 3 ·

Basic reporting

The manuscript passed all the criteria.

Experimental design

The manuscript passed all the criteria.

Validity of the findings

The manuscript passed all the criteria.